# An Axiomatic Characterization of Mutual Information

**DOI:** 10.3390/e25040663

**Published:** 2023-04-15

**Authors:** James Fullwood

**Affiliations:** School of Mathematical Sciences, Shanghai Jiao Tong University, Shanghai 200240, China; fullwood@sjtu.edu.cn

**Keywords:** shannon theory, information measures, mutual information

## Abstract

We characterize mutual information as the unique map on ordered pairs of discrete random variables satisfying a set of axioms similar to those of Faddeev’s characterization of the Shannon entropy. There is a new axiom in our characterization, however, which has no analog for Shannon entropy, based on the notion of a *Markov triangle*, which may be thought of as a composition of communication channels for which conditional entropy acts functorially. Our proofs are coordinate-free in the sense that no logarithms appear in our calculations.

## 1. Introduction

Axiomatic characterizations of information measures go back to the seminal work of Shannon [1], providing conceptual insights into their meaning as well as justification for the analytic formulae involved in their definitions. Various characterizations for Shannon entropy, relative entropy, Renyi and Tsallis entropies, von Neumann and Segal entropies, quantum relative entropy, as well as other generalized information measures have appeared in the literature [2,3,4,5,6,7,8,9], and a review of such enterprises in the classical (i.e., non-quantum) setting appears in the survey of Csiszár [10]. More recently, functorial characterizations of information measures from a categorical viewpoint have appeared in the works of Baez, Fritz, and Leinster [11,12], as well as our work with Parzygnat [13], who have proven a functorial characterization of the von Neumann entropy [14]. An axiomatic approach to entropy in the theory of biodiversity is the subject of the recent book [15] by Leinster.

In spite of the breadth of the aforementioned results, the mutual information of a pair of random variables seems to be missing from the story. While an operational characterization of mutual information in the context of algorithmic information theory appears in [16], to the best of our knowledge, an axiomatic characterization in the vein of those surveyed by Csiszár in [10] is absent in the literature. It is then the goal of the present work to introduce mutual information into the axiomatic framework.

Our main result is Theorem 1, where we prove that the mutual information I(X,Y) of an ordered pair of random variables is the unique function (up to an arbitrary multiplicative factor) on pairs of random variables satisfying the following axioms:**Continuity**: If $\left(X_{n},Y_{n}\right)\to \left(X,Y\right)$, then $I\left(X,Y\right)=\lim_{n\to \infty }I\left(X_{n},Y_{n}\right)$.**Strong Additivity:** Given a random variable X:Ω→X with probability mass function p:X→[0,1], and a collection of pairs of random variables $\left(Y^{x},Z^{x}\right)$ indexed by X, then
$$I\left(\underset{x\in X}{⨁}p\left(x\right)\left(Y^{x},Z^{x}\right)\right)=I\left(X,X\right)+\sum_{x\in X}p\left(x\right)I\left(Y^{x},Z^{x}\right).$$**Symmetry:**I(X,Y)=I(Y,X) for every pair of random variables (X,Y).**Invariance Under Pullbacks:** If $\pi :\Omega^{\prime }\to \Omega$ is a measure-preserving function, then for every pair of random variables (X,Y) with common domain Ω,
I(X,Y)=I(X∘π,Y∘π).**Weak Functoriality:** For every Markov triangle (X,Y,Z),
I(X,Z)=I(X,Y)+I(Y,Z)−I(Y,Y).**Vacuity:** If *C* is a constant random variable, then I(X,C)=0.

The fact that mutual information satisfies Axioms 1, 3, and 6 is well known to anybody familiar with mutual information. As we work at the level of random variables as opposed to simple probability distributions (which we do for wider applicability of our results), Axiom 4 is a reflection of the fact that mutual information only depends on probabilities. For Axiom 2, we define a convex structure on pairs of random variables in such a way that the strong additivity of Shannon entropy is generalized to our context. Axiom 5 is defined in terms of the notion of a *Markov triangle*, a concept we define based on the notion of a “coalescable” composition of communication channels which was introduced in [13]. Intuitively, a Markov triangle may be thought of as a composition of noisy channels over which the associated conditional entropy is additive. Moreover, such axioms are sharp in the sense that if any of the axioms are removed, then mutual information may not be characterized. In particular, the joint entropy H(X,Y) satisfies Axioms 1–5, while the conditional entropy H(Y|X) satisfies all the axioms except the symmetry Axiom 3 (note that since H(X,X)=0, Axiom 2 in the case of conditional entropy becomes convex linearity).

In the spirit of the axiomatic approach, we note that logarithms are absent from all calculations in this paper.

## 2. Mutual Information

Let (Ω,Σ,μ) be a probability space, where Ω is thought of as the set of all possible outcomes of a data generating process or experiment, Σ is a σ-algebra of measurable subsets of Ω, and μ is a probability measure.

**Definition** **1.***A*
 **finite random variable** 
*is a surjective function*
X:Ω→X
*such that*
X
*is a finite set and*
$X^{-1}\left(x\right)\in \Sigma$ for all x∈X. *In such a case, the set*
X
*is often referred to as the* **support***, or* **alphabet** *associated with* X. *The* **probability mass function** *of* X *is the function p:X→[0,1] given by*
$$p\left(x\right)=\mu \left(X^{-1}\left(x\right)\right),$$
*and the* **Shannon entropy** *of* *X is the non-negative real number H(X) given by*
$$H\left(X\right)=-\sum_{x\in X}p\left(x\right)\log\left(p(x)\right).$$*The collection of all finite random variables on* Ω *will be denoted FRV(Ω).*

**Definition** **2.**
*Let $\left(X,Y\right)\in \mathrm{FRV}\left(\Omega \right)\times \mathrm{FRV}\left(\Omega \right)$ be an ordered pair of random variables with supports X and Y respectively.*

*The* **joint distribution function** *of (X,Y) is the function ϑ:X×Y→[0,1] given by*$$\vartheta \left(x,y\right)=\mu \left(X^{-1}\left(x\right)\cap Y^{-1}\left(y\right)\right),$$*The* **joint entropy** *of (X,Y) is the non-negative real number given by*$$H\left(X,Y\right)=-\sum_{x\in X}\sum_{y\in Y}\vartheta \left(x,y\right)\log\left(\vartheta (x,y)\right),$$*The* **mutual information** *of (X,Y) is the real number I(X,Y) given by*I(X,Y)=H(X)+H(Y)−H(X,Y).


**Remark** **1.***With every pair of random variables (X,Y) one may associate a probability transition matrix p(y|x) given by*$$p\left(y|x\right)=\frac{\vartheta (x,y)}{p(x)},$$*where p:X→[0,1] is the probability mass function of* X. *As such, one may view (X,Y) as a noisy channel X⇝Y together with the prior distribution *p* on its set of inputs.*

We now list some well-known properties of mutual information which will be useful for our purposes (see, e.g., [17] for proofs).

**Proposition** **1.**
*Mutual information satisfies the following properties.*

*i*.
*I(X,Y)≥0 for all (X,Y)∈FRV(Ω)×FRV(Ω).*
*ii*.
*I(X,Y)=I(Y,X) for all (X,Y)∈FRV(Ω)×FRV(Ω).*
*iii*.
*I(X,X)=H(X) for all X∈FRV(Ω).*
*iv*.
*I(X,C)=0 for every constant random variable C∈FRV(Ω).*



**Definition** **3.***The* **canonical product** *on FRV(Ω) is the map P:FRV(Ω)×FRV(Ω)→FRV(Ω), given by P(X,Y)(ω)=(X(ω),Y(ω))∈X×Y for all ω∈Ω.*

**Proposition** **2.**
*Let (X,Y)∈FRV(Ω)×FRV(Ω). Then the following statements hold.*

*i*.
*The probability mass function of P(X,Y) is the joint distribution function ϑ(x,y). In particular, $H\left(X,Y\right)=H\left(P(X,Y)\right)$.*
*ii*.
*$I\left(X,P(X,Y)\right)=H\left(X\right)$.*



**Proof.** 
i.Let ν:X×Y→[0,1] denote the probability mass function of P(X,Y). Then for all (x,y)∈X×Y we have $P{\left(X,Y\right)}^{-1}\left(x,y\right)=X^{-1}\left(x\right)\cap Y^{-1}\left(y\right)$, thus
$$\nu \left(x,y\right)=\mu \left(P{\left(X,Y\right)}^{-1}\left(x,y\right)\right)=\mu \left(X^{-1}\left(x\right)\cap Y^{-1}\left(y\right)\right)=\vartheta \left(x,y\right),$$
as desired.ii.The statement follows from the fact that $H\left(X,P(X,Y)\right)=H\left(X,Y\right)$. □


## 3. Convexity

We now generalize the notion of a convex combination of probability distributions to the setting of pairs of random variables, which will be used to extend the notion of strong additivity for Shannon entropy to mutual information.

**Notation** **1.***We use the notation X∐Y to denote the disjoint union of the sets* X *and* Y.

**Definition** **4.***Let X be a finite set, and let p:X→[0,1] be a probability distribution on X. Then ${⨁}_{x\in X}p\left(x\right)\left(\Omega ,\Sigma ,\mu \right)$ is the probability space associated with the triple $\left(X\times \Omega ,X\times \Sigma ,p\times \mu \right)$. Now suppose $Y^{x}\in \mathrm{FRV}\left(\Omega \right)$ is a collection of random variables indexed by X, and let $q^{x}:Y^{x}\to \left[0,1\right]$ denote the probability mass function of $Y^{x}$. The* p**-weighted convex sum** 
*${⨁}_{x\in X}p\left(x\right)Y^{x}\in \mathrm{FRV}\left(X\times \Omega \right)$ is the random variable given by*
(1)$$\left(\underset{x\in X}{⨁}p\left(x\right)Y^{x}\right)\left(\tilde{x},\omega \right)=Y^{\tilde{x}}\left(\omega \right).$$
*It then follows that the probability mass function of ${⨁}_{x\in X}p\left(x\right)Y^{x}$ is a function of the form $r:{∐}_{x\in X}Y^{x}\to \left[0,1\right]$, and using the fact that ${∐}_{x\in X}Y^{x}$ is canonically isomorphic to the set*

$$\left\{\left(x,y\right)\;|\;x\in X\;\mathrm{and}\;y\in Y^{x}\right\},$$

*it follows that *r* is then given by $r\left(x,y\right)=p\left(x\right)q^{x}\left(y\right)$.*


A reformulation of the strong additivity property for Shannon entropy in terms of the convex structure just introduced for random variables is given using the following proposition.

**Proposition** **3.***Let X be a finite set, let p:X→[0,1] be a probability distribution on X, and suppose $Y^{x}\in \mathrm{FRV}\left(\Omega \right)$ is a collection of random variables indexed by X. Then*(2)$$H\left(\underset{x\in X}{⨁}p\left(x\right)Y^{x}\right)=H\left(p\right)+\sum_{x\in X}p\left(x\right)H\left(Y^{x}\right),$$*where H(p) is the Shannon entropy of the probability distribution *p.

**Proposition** **4.**
*Let X be a finite set, let p:X→[0,1] be a probability distribution on X, and suppose $\left(Y^{x},Z^{x}\right)\in \mathrm{FRV}\left(\Omega \right)\times \mathrm{FRV}\left(\Omega \right)$ is a collection of pairs of random variables indexed by X. Then*

(3)
$$\underset{x\in X}{⨁}p\left(x\right)P\left(Y^{x},Z^{x}\right)=P\left(\underset{x\in X}{⨁}p\left(x\right)Y^{x},\underset{x\in X}{⨁}p\left(x\right)Z^{x}\right)$$



**Proof.** Let $(\tilde{x},\omega )\in X\times \Omega$. Then
$$\begin{array}{ccc} \left(\underset{x\in X}{⨁}p\left(x\right)P\left(Y^{x},Z^{x}\right)\right)\left(\tilde{x},\omega \right)\overset{\left(1\right)}{=}P\left(Y^{\tilde{x}},Z^{\tilde{x}}\right)\left(\omega \right) & = & \left(Y^{\tilde{x}}\left(\omega \right),Z^{\tilde{x}}\left(\omega \right)\right)\\ & \overset{\left(1\right)}{=} & \left(\underset{x\in X}{⨁}p\left(x\right)Y^{x}\left(\tilde{x},\omega \right),\underset{x\in X}{⨁}p\left(x\right)Z^{x}\left(\tilde{x},\omega \right)\right)\\ & = & P\left(\underset{x\in X}{⨁}p\left(x\right)Y^{x},\underset{x\in X}{⨁}p\left(x\right)Z^{x}\right)\left(\tilde{x},\omega \right), \end{array}$$
thus Equation (Equation 3) holds. □

In light of Proposition 4, we make the following definition.

**Definition** **5.***Let X be a finite set, let p:X→[0,1] be a probability distribution on X, and suppose $\left(Y^{x},Z^{x}\right)\in \mathrm{FRV}\left(\Omega \right)\times \mathrm{FRV}\left(\Omega \right)$ is a collection of pairs of random variables indexed by X. The* p**-weighted convex sum** *${⨁}_{x\in X}p\left(x\right)\left(Y^{x},Z^{x}\right)\in \mathrm{FRV}\left(X\times \Omega \right)\times \mathrm{FRV}\left(X\times \Omega \right)$ is defined to be the the ordered pair $\left({⨁}_{x\in X}p\left(x\right)Y^{x},{⨁}_{x\in X}p\left(x\right)Z^{x}\right)$.*

**Proposition** **5**(Strong Additivity of Mutual Information). *Let X be a finite set, let p:X→[0,1] be a probability distribution on X, and suppose $\left(Y^{x},Z^{x}\right)\in \mathrm{FRV}\left(\Omega \right)\times \mathrm{FRV}\left(\Omega \right)$ is a collection of pairs of random variables indexed by X. Then*
$$I\left(\underset{x\in X}{⨁}p\left(x\right)\left(Y^{x},Z^{x}\right)\right)=H\left(p\right)+\sum_{x\in X}p\left(x\right)I\left(Y^{x},Z^{x}\right),$$
*where H(p) is the Shannon entropy of the probability distribution *p.

**Proof.** Indeed,
$$\begin{array}{ccc} I\left(\underset{x\in X}{⨁}p\left(x\right)\left(Y^{x},Z^{x}\right)\right) & = & I\left(\underset{x\in X}{⨁}p\left(x\right)Y^{x},\underset{x\in X}{⨁}p\left(x\right)Z^{x}\right)\\ & = & H\left(\underset{x\in X}{⨁}p\left(x\right)Y^{x}\right)+H\left(\underset{x\in X}{⨁}p\left(x\right)Z^{x}\right)-H\left(\underset{x\in X}{⨁}p\left(x\right)Y^{x},\underset{x\in X}{⨁}p\left(x\right)Z^{x}\right)\\ & \overset{\left(3\right)}{=} & H\left(\underset{x\in X}{⨁}p\left(x\right)Y^{x}\right)+H\left(\underset{x\in X}{⨁}p\left(x\right)Z^{x}\right)-H\left(\underset{x\in X}{⨁}p\left(x\right)P\left(Y^{x},Z^{x}\right)\right)\\ & \overset{\left(2\right)}{=} & 2H\left(p\right)+\sum_{x\in X}p\left(x\right)\left(H\left(Y^{x}\right)+H\left(Z^{x}\right)\right)-\left(H\left(p\right)+\sum_{x\in X}p\left(x\right)H\left(Y^{x},Z^{x}\right)\right)\\ & = & H\left(p\right)+\sum_{x\in X}p\left(x\right)\left(H\left(Y^{x}\right)+H\left(Z^{x}\right)-H\left(Y^{x},Z^{x}\right)\right)\\ & = & H\left(p\right)+\sum_{x\in X}p\left(x\right)I\left(Y^{x},Z^{x}\right), \end{array}$$
as desired. □

## 4. Continuity

**Definition** **6.***Let $X_{n}\in \mathrm{FRV}\left(\Omega \right)$ be a sequence of random variables, and let $p_{n}:X_{n}\to \left[0,1\right]$ be the associated sequence of probability mass functions. Then $X_{n}$ is said to* **weakly converge** *(or* **converge in distribution***) to the random variable X∈FRV(Ω) with probability mass function p:X→[0,1] if the following conditions hold.*
*i*.
*There exists an N∈N for which $X_{n}=X$ for all n≥N.*
*ii*.
*For all x∈X we have $\lim_{n\to \infty }p_{n}\left(x\right)=p\left(x\right)$, i.e., $p_{n}\to p$ pointwise.*

*In such a case, we write $X_{n}\to X$. If $\left(X_{n},Y_{n}\right)\in \mathrm{FRV}\left(\Omega \right)\times \mathrm{FRV}\left(\Omega \right)$ is a sequence of pairs of random variables, then $\left(X_{n},Y_{n}\right)$ is said to* **weakly converge** *to (X,Y)∈FRV(Ω)×FRV(Ω) if $P\left(X_{n},Y_{n}\right)\to P\left(X,Y\right)$.*

**Proposition** **6.**
*Shannon entropy is continuous, i.e., if $X_{n}\to X$, then*

$$H\left(X\right)=\lim_{n\to \infty }H\left(X_{n}\right).$$



**Proof.** This result is standard, see, e.g., [3] or [11]. □

**Proposition** **7.**
*Mutual information is continuous, i.e., if $\left(X_{n},Y_{n}\right)\to \left(X,Y\right)$, then*

$$I\left(X,Y\right)=\lim_{n\to \infty }I\left(X_{n},Y_{n}\right).$$



**Proof.** Suppose $\left(X_{n},Y_{n}\right)\to \left(X,Y\right)$, so that $X_{n}\to X$, $Y_{n}\to Y$ and $H\left(X_{n},Y_{n}\right)\to H\left(X,Y\right)$. We then have
$$I\left(X,Y\right)=H\left(X\right)+H\left(Y\right)-H\left(X,Y\right)=\lim_{n\to \infty }\left(H\left(X_{n}\right)+H\left(Y_{n}\right)-H\left(X_{n},Y_{n}\right)\right)=\lim_{n\to \infty }I\left(X_{n},Y_{n}\right),$$
as desired. □

## 5. Markov Triangles

In this section, we define the notion of a *Markov triangle*, a concept based on the notion of a “coalescable” composition of communication channels which was introduced in [13]. Such a notion will be crucial for our characterization of mutual information.

**Definition** **7.***Let X∈FRV(Ω) be a random variable with probability mass function p:X→[0,1], and let x∈X. Then for any random variable Y∈FRV(Ω), the* **conditional distribution function** *of Y given X=x is the function $q^{x}:Y\to \left[0,1\right]$ given by*$$q^{x}\left(y\right)=\left\{\begin{array}{c} \frac{\vartheta (x,y)}{p(x)}\;\mathrm{if}\;p\left(x\right)\neq 0\\ 0\;\mathrm{otherwise}. \end{array}\right.$$*From here on, the value $q^{x}\left(y\right)$ will be denoted q(y|x). The* **conditional entropy** *of* Y *given* X *is the non-negative real number H(Y|X) given by*
$$H\left(Y|X\right)=\sum_{x\in X}p\left(x\right)H\left(q^{x}\right),$$
*where $H(q^{x})$ is the Shannon entropy of the distribution $q^{x}$ on *Y.

**Proposition** **8.**
*Let (X,Y) be a pair of random variables. Then*

(4)
I(X,Y)=I(Y,Y)−H(Y|X).



**Proof.** Since I(Y,Y)=H(Y), the statement follows from the well-known fact that I(X,Y)=H(Y)−H(Y|X), the proof of which may be found in any information theory text (e.g., [17]). □

**Definition** **8.***Let (X,Y,Z) be a triple of random variables with supports X, Y and Z respectively, and let q(y|x), p(z|y) and r(z|x) denote the associated conditional distribution functions. Then (X,Y,Z) is said to form a* **Markov triangle** *if there exists a function h:Z×X→Y such that for all (z,x)∈Z×X we have*$$r\left(z|x\right)=p\left(z|h(z,x)\right)q\left(h(z,x)|x\right).$$*In such a case, *h* is said to be a* **mediator function** *for the triple (X,Y,Z).*

**Remark** **2.***A Markov triangle (X,Y,Z) with supports X, Y, and Z may be thought of as a composition of noisy channels $X\overset{f}{⇝}Y\overset{g}{⇝}Z$ such that if z∈Z is the output of the channel g∘f, and one is given the information that the associated input was x∈X, then the output at the intermediary stage Y was necessarily y=h(z,x) (where* h *is the associated mediator function). In other words, if P is a symbol for general probabilities, then*
P(x,z|y)=P(x|y)P(z|y),
*thus the Markov triangle condition says that* X *and* Z *are conditionally independent given Y. As compositions of deterministic channels always satisfy this property, Markov triangles are a generalization of compositions of deterministic channels. While Markov triangles play a crucial role in our characterization of mutual information and also the characterizations of conditional entropy and information loss in [13], their broader significance in the study of information measures has yet to be determined.*

**Proposition** **9.**
*Suppose (X,Y,Z) is a Markov triangle. Then*

I(X,Z)=I(X,Y)+I(Y,Z)−I(Y,Y).

*In particular, I(X,Z)≤I(X,Y)+I(Y,Z).*


Before giving a proof of Proposition 9, we first need the following lemma.

**Lemma** **1.**
*Suppose (X,Y,Z) is a Markov triangle. Then*

(5)
H(Z|X)=H(Z|Y)+H(Y|X).



**Proof.** The statement is simply a reformulation of Theorem 2 in [13]. □

**Proof** **of** **Proposition** **9.**Suppose (X,Y,Z) is a Markov triangle. Then
$$\begin{array}{ccc} I\left(X,Z\right)\overset{\left(4\right)}{=}I\left(Z,Z\right)-H\left(Z|X\right) & \overset{\left(5\right)}{=} & I\left(Z,Z\right)-\left(H(Z|Y)+H(Y|X)\right)\\ & = & I(Y,Y)-H(Y|X)+I(Z,Z)-H(Z|Y)-I(Y,Y)\\ & \overset{\left(4\right)}{=} & I(X,Y)+I(Y,Z)-I(Y,Y), \end{array}$$
as desired. □

**Proposition** **10.**
*Let X,Y∈FRV(Ω) be random variables with probability mass functions p:X→[0,1] and q:Y→[0,1] respectively. Then the following statements hold.*

*i*.
*The triple $\left(X,P(X,Y),Y\right)$ is a Markov triangle.*
*ii*.
*If $f:X\to X^{\prime }$ is a bijection, then the triple (X,f∘X,Y) is a Markov triangle.*
*iii*.
*If $g:Y\to Y^{\prime }$ is a bijection, then the triple (X,Y,g∘Y) is a Markov triangle.*



**Proof.** 
i.Let r(y|x) be the conditional distribution associated with (X,Y), let $p\left(y|(\tilde{x},\tilde{y})\right)$ be the conditional distribution associated with $\left(P(X,Y),Y\right)$, and let $q\left((\tilde{x},\tilde{y})|x\right)$ be the conditional distribution function associated with $\left(X,P(X,Y)\right)$. Then for all y∈Y and x∈X we have
$$r\left(y|x\right)=\sum_{(\tilde{x},\tilde{y})\in X\times Y}p\left(y|(\tilde{x},\tilde{y})\right)q\left((\tilde{x},\tilde{y})|x\right)=p\left(y|(x,y)\right)q\left((x,y)|x\right),$$
where the second equality comes from the fact that $p\left(y|(\tilde{x},\tilde{y})\right)=0$ unless $\tilde{y}=y$ and $q\left((\tilde{x},\tilde{y})|x\right)=0$ unless $x=\tilde{x}$. It then follows that the function h:Y×X→X×Y given by h(y,x)=(x,y) is a mediator function for $\left(X,P(X,Y),Y\right)$, thus $\left(X,P(X,Y),Y\right)$ is a Markov triangle.ii.Let r(y|x) be the conditional distribution associated with (X,Y), let $p\left(y|x^{\prime }\right)$ be the conditional distribution associated with $\left(f\circ X,Y\right)$, and let $q\left(x^{\prime }|x\right)$ be the conditional distribution function associated with $\left(X,f\circ X\right)$. Then for all y∈Y and x∈X we have
$$r\left(y|x\right)=\sum_{x^{\prime }\in X^{\prime }}p\left(y|x^{\prime }\right)q\left(x^{\prime }|x\right)=p\left(y|f(x)\right)q\left(f(x)|x\right),$$
where the second equality comes from the fact that $q(x^{\prime }|x)=0$ unless $x^{\prime }=f\left(x\right)$. It then follows that the function $h:Y\times X\to X^{\prime }$ given by h(y,x)=f(x) is a mediator function for (X,f∘X,Y), thus (X,f∘X,Y) is a Markov triangle.iii.Let $r(y^{\prime }|x)$ be the conditional distribution associated with (X,g∘Y), let $p\left(y^{\prime }|y\right)$ be the conditional distribution associated with $\left(Y,g\circ Y\right)$, and let $q\left(y|x\right)$ be the conditional distribution associated with $\left(X,Y\right)$. Then for all $y^{\prime }\in Y^{\prime }$ and x∈X we have
$$r\left(y^{\prime }|x\right)=\sum_{y\in Y}p\left(y^{\prime }|y\right)q\left(y|x\right)=p\left(y^{\prime }|g^{-1}\left(y^{\prime }\right)\right)q\left(g^{-1}\left(y^{\prime }\right)|x\right),$$
where the second equality comes from the fact that $p(y^{\prime }|y)=0$ unless $y=g^{-1}\left(y^{\prime }\right)$. It then follows that the function $h:Y^{\prime }\times X\to X^{\prime }$ given by $h\left(y^{\prime },x\right)=g^{-1}\left(y^{\prime }\right)$ is a mediator function for (X,Y,g∘Y), thus (X,Y,g∘Y) is a Markov triangle. □


## 6. Characterization Theorem

We now state and prove our characterization theorem for mutual information.

**Definition** **9.***Let (Ω,Σ,μ) and $\left(\Omega^{\prime },\Sigma^{\prime },\mu^{\prime }\right)$ be probability spaces. A map $\pi :\Omega^{\prime }\to \Omega$ is said to be* **measure-preserving** *if for all σ∈Σ we have $\pi^{-1}\left(\sigma \right)\in \Sigma^{\prime }$ and*$$\mu^{\prime }\left(\pi^{-1}\left(\sigma \right)\right)=\mu \left(\sigma \right).$$

**Definition** **10.***Let* F *be a map that sends pairs of random variables to the real numbers.*
F *is said to be* **continuous** *if*
(6)$$F\left(X,Y\right)=\lim_{n\to \infty }F\left(X_{n},Y_{n}\right)$$
*whenever $\left(X_{n},Y_{n}\right)\to \left(X,Y\right)$.*F *is said to be* **strongly additive** *if given a random variable* X *with probability mass function p:X→[0,1], and a collection of pairs of random variables $\left(Y^{x},Z^{x}\right)$ indexed by X, then*
(7)$$F\left(\underset{x\in X}{⨁}p\left(x\right)\left(Y^{x},Z^{x}\right)\right)=F\left(X,X\right)+\sum_{x\in X}p\left(x\right)F\left(Y^{x},Z^{x}\right).$$F *is said to be* **symmetric** *if F(X,Y)=F(Y,X) for every pair of random variables (X,Y).*F *is said to be* **invariant under pullbacks** *if for every pair of random variables (X,Y)∈FRV(Ω)×FRV(Ω) and every measure-preserving map $\pi :\Omega^{\prime }\to \Omega$ we have*
(8)F(X,Y)=F(X∘π,Y∘π).F *is said to be* **weakly functorial** *if for every Markov triangle (X,Y,Z) we have*
(9)F(X,Z)=F(X,Y)+F(Y,Z)−F(Y,Y).


**Remark** **3.***The terminology “weakly functorial” comes from viewing (Equation 9) from a category-theoretic perspective. In particular, with a pair of random variables (X,Y) one may associate a noisy channel $X\overset{f}{⇝}Y$ where X=Supp(X) and Y=Supp(Y), so that a Markov triangle (X,Y,Z) then corresponds to a composition $X\overset{f}{⇝}Y\overset{g}{⇝}Z$ with Z=Supp(Z). If FinPS denotes the category of noisy channels and BR denotes the category with one object whose morphisms are the real numbers (with a composition corresponding to addition), then a map F:FinPS→BR is a functor if*(10)F(g∘f)=F(g)+F(f).*Rewriting (Equation 10) in terms of the pairs of random variables for which the morphisms* f, g, *and g∘f are associated with, then the functoriality condition (Equation 10) reads*
F(X,Z)=F(X,Y)+F(Y,Z),
*thus the condition F(X,Z)≤F(X,Y)+F(Y,Z) is a weaker form of functoriality. For more on information measures from a category-theoretic perspective, see [11,12,13,14].*

**Theorem** **1**(Axiomatic Characterization of Mutual Information). *Let* F *be a map that sends pairs of random variables to the non-negative real numbers, and suppose *F *satisfies the following conditions.*
*1*.F *is continuous.**2*.F *is strongly additive.**3*.F *is symmetric.**4*.F *is weakly functorial.**5*.F *is invariant under pullbacks.**6*.
*F(X,C)=0 for every constant random variable *C*.*

*Then* F *is a non-negative multiple of mutual information. Conversely, mutual information satisfies conditions 1–6.*

Before giving a proof, we first need several lemmas. The first lemma states that a map F on pairs of random variables which is continuous and invariant under pullbacks, only depends on the underlying probability mass functions of the random variables.

**Lemma** **2.**
*Let F be a map from pairs of random variables to the real numbers, which is continuous and invariant under pullbacks, and suppose (X,Y)∈FRV(Ω)×FRV(Ω) and $\left(X^{\prime },Y^{\prime }\right)\in \mathrm{FRV}\left(\Omega^{\prime }\right)\times \mathrm{FRV}\left(\Omega^{\prime }\right)$ are such that the associated joint distribution functions ϑ:X×Y→[0,1] and $\vartheta^{\prime }:X\times Y\to \left[0,1\right]$ are equal. Then $F\left(X,Y\right)=F\left(X^{\prime },Y^{\prime }\right)$.*


**Proof.** Let $\pi :\Omega \times \Omega^{\prime }\to \Omega$ and $\pi^{\prime }:\Omega \times \Omega^{\prime }\to \Omega^{\prime }$ be the natural projections. Since both the natural projections are measure-preserving, we have $F\left(X,Y\right)=F\left(X\circ \pi ,Y\circ \pi \right)$, $F\left(X^{\prime },Y^{\prime }\right)=F\left(X^{\prime }\circ \pi^{\prime },Y^{\prime }\circ \pi^{\prime }\right)$, and moreover, from the assumption that $\vartheta =\vartheta^{\prime }$ it follows that the joint distribution functions associated with $\left(X\circ \pi ,Y\circ \pi \right)$ and $\left(X^{\prime }\circ \pi^{\prime },Y^{\prime }\circ \pi^{\prime }\right)$ are equal. It then follows that if $\left(X_{n},Y_{n}\right)$ is the constant sequence given by $X_{n}=X^{\prime }\circ \pi^{\prime }$ and $Y_{n}=Y^{\prime }\circ \pi^{\prime }$ for all n∈N, then $\left(X_{n},Y_{n}\right)\to \left(X\circ \pi ,Y\circ \pi \right)$ (since $P\left(X_{n},Y_{n}\right)\to P\left(X\circ \pi ,Y\circ \pi \right)$). We then have
$$F\left(X,Y\right)\overset{\left(8\right)}{=}F\left(X\circ \pi ,Y\circ \pi \right)\overset{\left(6\right)}{=}\lim_{n\to \infty }F\left(X_{n},Y_{n}\right)=F\left(X^{\prime }\circ \pi^{\prime },Y^{\prime }\circ \pi^{\prime }\right)\overset{\left(8\right)}{=}F\left(X^{\prime },Y^{\prime }\right),$$
as desired. □

**Lemma** **3.**
*Let X be a random variable with probability mass function p:X→[0,1], let f:X→Y be a bijection, and suppose *C* is a constant random variable. Then the following statements hold.*

*i*.
*The triples $\left(X,f\circ X,C\right)$ and $\left(f\circ X,X,C\right)$ are both Markov triangles.*
*ii*.
*Let *F* be a map that sends pairs of random variables to real numbers, and suppose *F* is symmetric and weakly functorial. Then*



(11)
$$F\left(X,C\right)-F\left(f\circ X,C\right)+F\left(f\circ X,f\circ X\right)=F\left(f\circ X,C\right)-F\left(X,C\right)+F\left(X,X\right).$$



**Proof.** 
i.The statement follows from item ii of Proposition 10.ii.By item 3, the triples $\left(X,f\circ X,C\right)$ and $\left(f\circ X,X,C\right)$ are both Markov triangles, thus the weak functoriality of F yields
(12)F(X,C)=F(X,f∘X)+F(f∘X,C)−F(f∘X,f∘X),
and
(13)F(f∘X,C)=F(f∘X,X)+F(X,C)−F(X,X).And since F is symmetric F(X,f∘X)=F(f∘X,X), thus Equations (Equation 12) and (Equation 13) imply Equation (Equation 11), as desired. □


The next lemma is Baez, Fritz, and Leinster’s reformulation of Faddeev’s characterization of Shannon entropy [3], which they use in their characterization of the information loss associated with a deterministic mapping [11]. This lemma will allow us to relate F(X,X) to the Shannon entropy H(X).

**Lemma** **4.**
*Let S be a map that sends finite probability distributions to the non-negative real numbers, and suppose S satisfies the following conditions.*

*i*.
*S is continuous, i.e., if $p_{n}:X\to \left[0,1\right]$ is a convergent sequence of probability distributions on a finite set X (i.e., if $\lim_{n\to \infty }p_{n}\left(x\right)$ exists for all x∈X), then*

$$S\left(\lim_{n\to \infty }p_{n}\right)=\lim_{n\to \infty }S\left(p_{n}\right).$$

*ii*.
*S(1)=0 for the distribution 1:{★}→[0,1].*
*iii*.
*If q:Y→[0,1] is a probability distribution on a finite set Y and f:X→Y is a bijection, then S(q)=S(q∘f).*
*iv*.
*If p:X→[0,1] is a probability distribution on a finite set X, and $q^{x}:Y^{x}\to \left[0,1\right]$ is a collection of finite probability distributions indexed by X, then*

$$S\left(\underset{x\in X}{⨁}p\left(x\right)q^{x}\right)=S\left(p\right)+\sum_{x\in X}p\left(x\right)S\left(q^{x}\right),$$

*where ${⨁}_{x\in X}p\left(x\right)q^{x}:{∐}_{x\in X}Y^{x}\to \left[0,1\right]$ is the finite distribution given by $\left({⨁}_{x\in X}p\left(x\right)q^{x}\right)$$\left(\tilde{x},y_{\tilde{x}}\right)=p\left(\tilde{x}\right)q^{\tilde{x}}\left(y_{\tilde{x}}\right)$.*


*Then S is a non-negative multiple of Shannon entropy.*


**Lemma** **5.***Let* F *be a map that sends pairs of random variables to the non-negative real numbers satisfying conditions 1–6 of Theorem 1, and let E be the map on random variables given by*
E(X)=F(X,X).
*Then E is a non-negative multiple of Shannon entropy.*


**Proof.** Let ϕ be the map that takes a random variable to its probability mass function, let σ be a section (so that ϕ∘σ is the identity), and let S=E∘σ. Since *F* is invariant under pullbacks (condition 5 of Theorem 1) Lemma 2 holds, thus the map S is independent of the choice of a section σ of ϕ, and as such, it follows that E=S∘ϕ. We now show that S satisfies items i–iv of Lemma 4, which then implies E(X) is a non-negative multiple of the Shannon entropy H(X).
i.Let $p_{n}:X\to \left[0,1\right]$ be a sequence of probability distributions on a finite set X, and suppose $\lim_{n\to \infty }p_{n}=p$. It then follows that $X_{n}=\sigma \left(p_{n}\right)$ weakly converges to X=σ(p), thus
$$\begin{array}{ccc} S\left(\lim_{n\to \infty }p_{n}\right) & = & S\left(p\right)=\left(E\circ \sigma \right)\left(p\right)=E\left(X\right)=F\left(X,X\right)=\lim_{n\to \infty }F\left(X_{n},X_{n}\right)\\ & = & \lim_{n\to \infty }E\left(X_{n}\right)=\lim_{n\to \infty }E\left(\sigma \left(p_{n}\right)\right)=\lim_{n\to \infty }S\left(p_{n}\right) \end{array}$$
where the fifth equality follows from the continuity assumption on F (condition 1 of Theorem 1).ii.Let 1:{★}→[0,1] be a point mass distribution, so that σ(1)=C with *C* a constant random variable. Then S(1)=E(σ(1))=E(C)=F(C,C)=0, where the last equality follows from condition 6 of Theorem 1, i.e., that F(X,C)=0 for every constant random variable C.iii.Let X be a random variable with probability mass function p:X→[0,1], and suppose f:X→Y is a bijection. Since *F* is symmetric and weakly functorial (conditions 3 and 4 of Theorem 1), the hypotheses of item ii Lemma 3 are satisfied, so that Equation (Equation 11) holds, i.e., for any constant random variable C we have
$$F\left(X,C\right)-F\left(f\circ X,C\right)+F\left(f\circ X,f\circ X\right)=F\left(f\circ X,C\right)-F\left(X,C\right)+F\left(X,X\right).$$And since $F\left(X,C\right)=F\left(f\circ X,C\right)=0$ by condition 6 of Theorem 1, it follows that F(X,X)=F(f∘X,f∘X). Now let q:Y→[0,1] be the probability mass function of f∘X, so that $q=p\circ f^{-1}$. We then have
$$S\left(p\right)=E\left(X\right)=F\left(X,X\right)=F\left(f\circ X,f\circ X\right)=E\left(f\circ X\right)=S\left(q\right)=S\left(p\circ f^{-1}\right),$$
thus S satisfies item iii of Faddeev’s Theorem.iv.Let X be a random variable with probability mass function p:X→[0,1], $Y^{x}$ a collection of random variables indexed by X, and let $q^{x}:Y^{x}\to \left[0,1\right]$ be the associated probability mass functions for all x∈X. Then ${⨁}_{x\in X}p\left(x\right)Y^{x}$ has probability mass function ${⨁}_{x\in X}p\left(x\right)q^{x}$, thus
$$\begin{array}{ccc} S\left(\underset{x\in X}{⨁}p\left(x\right)q^{x}\right)=E\left(\underset{x\in X}{⨁}p\left(x\right)Y^{x}\right) & = & F\left(\underset{x\in X}{⨁}p\left(x\right)Y^{x},\underset{x\in X}{⨁}p\left(x\right)Y^{x}\right)\\ & \overset{\left(3\right)}{=} & F\left(\underset{x\in X}{⨁}p\left(x\right)\left(Y^{x},Y^{x}\right)\right)\\ & = & F\left(X,X\right)+\sum_{x\in X}p\left(x\right)F\left(Y^{x},Y^{x}\right)\\ & = & E\left(X\right)+\sum_{x\in X}p\left(x\right)E\left(Y^{x}\right)\\ & = & S\left(p\right)+\sum_{x\in X}p\left(x\right)S\left(q^{x}\right), \end{array}$$
where the fourth equality follows from the strong additivity of F, i.e., condition 2 of Theorem 1. It then follows that S satisfies item iv of Faddeev’s Theorem, as desired. □


The next lemma is the analog of property iii of Lemma 4 for information measures on pairs of random variables.

**Lemma** **6.**
*Let X,Y∈FRV(Ω) be random variables with probability mass functions p:X→[0,1] and q:Y→[0,1] respectively, and suppose F is a map on pairs of random variables to the real numbers which is symmetric, weakly functorial, and F(X,C)=0 for every constant random variable *C*. If $f:X\to X^{\prime }$ and $g:Y\to Y^{\prime }$ are bijections, then*

(14)
$$F\left(X,Y\right)=F\left(f\circ X,g\circ Y\right).$$



**Proof.** Since *f* is a bijection, (X,f∘X,X) is a Markov triangle by item ii of Proposition 10, thus
(15)F(X,X)=F(X,f∘X)+F(f∘X,X)−F(f∘X,f∘X).From the proof of Lemma 5 it follows that if F is weakly functorial, symmetric and F(X,C)=0 for every constant random variable C, F(f∘X,f∘X)=F(X,X). Moreover by the symmetry of *F* we have F(X,f∘X)=F(f∘X,X), thus Equation (Equation 15) implies F(X,X)=F(X,f∘X).Now consider the triples (f∘X,X,g∘Y) and (X,Y,g∘Y), which are both Markov triangles by items ii and iii of Proposition 10. The weakly functorial assumption on F then yields
$$\begin{array}{ccc} F\left(f\circ X,g\circ Y\right) & = & F\left(f\circ X,X\right)+F\left(X,g\circ Y\right)-F\left(X,X\right)\\ & = & F\left(f\circ X,X\right)+\left(F\left(X,Y\right)+F\left(Y,g\circ Y\right)-F\left(Y,Y\right)\right)-F\left(X,X\right), \end{array}$$
and since $F\left(f\circ X,X\right)=F\left(X,X\right)$ and $F\left(Y,g\circ Y\right)=F\left(Y,Y\right)$, it follows that $F\left(X,Y\right)=F\left(f\circ X,g\circ Y\right)$, as desired. □

The next lemma, together with the fact that (X,P(X,Y),Y) is a Markov triangle (by Proposition 10) is the crux of the proof, as we will soon see.

**Lemma** **7.**
*Let F be a map from pairs of random variables to the real numbers satisfying conditions 1–6 of Theorem 1, and let (X,Y) be a pair of random variables. Then $F\left(X,P(X,Y)\right)=F\left(X,X\right)$ and $F\left(P(X,Y),Y\right)=F\left(Y,Y\right)$.*


**Proof.** Let p:X→[0,1] and q:Y→[0,1] be the probability mass functions of X and Y respectively, and for all x∈X, let $Y^{x}$ be a random variable with probability mass function $q^{x}:Y\to \left[0,1\right]$ given by $q^{x}\left(y\right)=q\left(y|x\right)$, so that $q^{x}$ is the conditional distribution of Y given X=x. By pulling back to larger sample spaces if necessary, we can assume without loss of generality that each $Y^{x}\in FRV\left(\Omega \right)$ for some fixed Ω. We also let $C^{x}\in FRV\left(\Omega \right)$ be the constant random variable supported on {x} for all x∈X, we let $f:{∐}_{x\in X}\left\{x\right\}\to X$ and $g:{∐}_{x\in X}Y\to X\times Y$ be the canonical bijections, and we let π:X×Ω→Ω be the natural projection. It then follows that $f\circ {⨁}_{x\in X}p\left(x\right)C^{x}$ and X∘π both have probability mass function p:X→[0,1], and also, that $g\circ {⨁}_{x\in X}p\left(x\right)Y^{x}$ and P(X,Y)∘π both have probability mass function equal to the joint distribution function ϑ:X×Y→[0,1] associated with (X,Y), thus Lemma 2 yields
(16)$$F\left(X\circ \pi ,P(X,Y)\circ \pi \right)=F\left(f\circ \underset{x\in X}{⨁}p\left(x\right)C^{x},g\circ \underset{x\in X}{⨁}p\left(x\right)Y^{x}\right).$$We then have
$$\begin{array}{ccc} F\left(X,P(X,Y)\right)\overset{\left(8\right)}{=}F\left(X\circ \pi ,P(X,Y)\circ \pi \right) & \overset{\left(16\right)}{=} & F\left(f\circ \underset{x\in X}{⨁}p\left(x\right)C^{x},g\circ \underset{x\in X}{⨁}p\left(x\right)Y^{x}\right)\\ & \overset{\left(14\right)}{=} & F\left(\underset{x\in X}{⨁}p\left(x\right)C^{x},\underset{x\in X}{⨁}p\left(x\right)Y^{x}\right)\\ & \overset{\left(3\right)}{=} & F\left(\underset{x\in X}{⨁}p\left(x\right)\left(C^{x},Y^{x}\right)\right)\\ & \overset{\left(7\right)}{=} & F\left(X,X\right)+\sum_{x\in X}p\left(x\right)F\left(C^{x},Y^{x}\right)\\ & = & F(X,X), \end{array}$$
where the last equality follows from the fact that F(C,X)=0 for every constant random variable C (since F is symmetric and F(X,C)=0 for every constant random variable C).As for $F\left(P(X,Y),Y\right)$, first note that $F\left(Y,P(Y,X)\right)=F\left(Y,Y\right)$ by what what we have just proved. We then have
$$F\left(P(X,Y),Y\right)=F\left(Y,P(X,Y)\right)=F\left(Y,P(Y,X)\right)=F\left(Y,Y\right),$$
where the first and second equalities follow from symmetry and invariance under pullbacks. □

**Proof of Theorem 1** Suppose F is a map from pairs of random variables to the non-negative real numbers satisfying conditions 1–6 of Theorem 1. According to Lemma 5, there exists a constant c≥0 such that F(X,X)=cH(X) for all random variables X. Now let (X,Y) be an arbitrary pair of random variables. According to Proposition 10, the triple (X,P(X,Y),Y) is a Markov triangle, thus
$$\begin{array}{ccc} F(X,Y) & \overset{\left(9\right)}{=} & F\left(X,P(X,Y)\right)+F\left(P(X,Y),Y\right)-F\left(P(X,Y),P(X,Y)\right)\\ & = & F\left(X,X\right)+F\left(Y,Y\right)-cH\left(P(X,Y)\right)\\ & = & cH(X)+cH(Y)-cH(X,Y)\\ & = & cI(X,Y), \end{array}$$
where the second equality follows from Lemma 7 and Lemma 5, and the third equality follows from Lemma 5 and item 2 of Proposition 2, thus F is a non-negative multiple of mutual information.Conversely, mutual information satisfies condition 1 by Proposition 7, condition 2 by Proposition 5, condition 3 by item 1 of Proposition 1, condition 4 by Proposition 9, condition 5 by the fact that mutual information only depends on probabilities, and condition 6 by item 1 of Proposition 1. □

## Data Availability

Not applicable.

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
