# Peer review of "An Axiomatic Characterization of Mutual Information"

_entropy, 2023, doi:10.3390/e25040663_

Round 1

Reviewer 1 Report

 1. On page 1, "Beaz" should be "Baez"
2. Theorem 2.6 ii. seems incorrect. Since the double sum in the definition of H is over the two sets, the joint entropy of X and the canonical product should be a multiple of the joint entropy of X and Y. The multiplier would be |X|. It looks like 2.6 ii. is not used elsewhere in the paper.

Author Response

Dear Reviewer,

Thank you for your report. Theorem 2.6 ii is correct. While there is a double sum involved, the conditional probabilities associated with the pair of random variables (X,P(X,Y)) are of the form

p((x,y)|x'),

where x' is an element of the support of X and (x,y) is an element of the support of P(X,Y). But this conditional probability is 0 unless x=x', and this is why a factor of |X| does not appear.

I have changed Beaz to Baez in the intro.

With kind regards,

James Fullwood 

Reviewer 2 Report

This work has clear logic, reasonable structure and sufficient proof. It is therefore acceptable in its current form

Author Response

Dear Reviewer,

Thank you for your report.

With kind regards,

James Fullwood 

Author Response

Dear Reviewer,

Thank you for your report. As for you comments 1, 2 and 4, I think perhaps we use different conventions. What you call the "distribution function" of a random variable I would call the "cumulative distribution function".

As for your comment 3, while it is more customary to throw away the random variables in information theory and just work with probability distributions, in this paper I keep the random variables around. So Definition 3.1 is just the analogue of taking a convex combination of probability distributions, but at the level of random variables. Moreover, the probability function associated with this construction is not the joint distribution of (X,Y) for some random variable Y. This only happens when Y^x=Y for all x in X. In particular, the variable y in the formula I give r(x,y)=p(x)q^x(y) actually depends on x, since in this case y must be an element of Y^x. As for the symbol \coprod, it means "coproduct", which in the category of sets just means disjoint union. I have added "Notation 3.1" to make the notation more clear. 

You comment 5 was very helpful, and makes the notion of Markov triangle indeed more clear. I have incorporated this comment into Remark 5.5.

I have corrected the typo "cannel" on page 10.

With kind regards,

James Fullwood